# Sugar promotes vegetative phase change in *Arabidopsis thaliana* by repressing the expression of *MIR156A* and *MIR156C*

Li Yang, Mingli Xu, Yeonjong Koo, Jia He, R Scott Poethig*

Department of Biology, University of Pennsylvania, Philadelphia, United States

**Abstract** Nutrients shape the growth, maturation, and aging of plants and animals. In plants, the juvenile to adult transition (vegetative phase change) is initiated by a decrease in miR156. In *Arabidopsis*, we found that exogenous sugar decreased the abundance of miR156, whereas reduced photosynthesis increased the level of this miRNA. This effect was correlated with a change in the timing of vegetative phase change, and was primarily attributable to a change in the expression of two genes, *MIR156A* and *MIR156C*, which were found to play dominant roles in this transition. The glucose-induced repression of miR156 was dependent on the signaling activity of HEXOKINASE1. We also show that the defoliation-induced increase in miR156 levels can be suppressed by exogenous glucose. These results provide a molecular link between nutrient availability and developmental timing in plants, and suggest that sugar is a component of the leaf signal that mediates vegetative phase change.

## Introduction

Higher plants and animals undergo several distinct transitions during their post-embryonic development. The correct timing of these developmental events is critical for survival and reproduction, and is regulated by complex interactions between endogenous and environmental factors, such as nutrition, temperature, or photoperiod (*Bernier, 1988*; *Rougvie, 2005*; *Amasino, 2010*; *Tolson and Chappell, 2012*). In plants, the maturation of the shoot can be divided into three phases: a juvenile vegetative phase, an adult vegetative phase, and a reproductive phase (*Poethig, 1990*). The juvenile to adult transition (vegetative phase change) is regulated by miR156 (*Wu and Poethig, 2006*; *Chuck et al., 2007*), a microRNA that is conserved throughout the plant kingdom (*Axtell and Bowman, 2008*). miR156 is highly expressed in the juvenile phase and decreases dramatically during vegetative phase change. This decrease produces an increase in the expression of its direct targets, SQUAMOSA PROMOTER BINDING PROTEIN-LIKE (SBP/SPL) transcription factors (*Schwab et al., 2005*; *Wu et al., 2009*), which go on to mediate the morphological and physiological changes associated with this transition. The basis for the decrease in miR156 expression is unknown.

One of the most obvious signs of phase change is a change in leaf shape. Goebel termed this phenomenon 'heteroblasty', and recognized that shoot development could be divided into more-or-less discrete stages on the basis of this trait (*Goebel, 1889*, *1900*). Goebel believed that the transition between juvenile and adult leaf types—the hallmark of vegetative phase change—is the result of a change in the nutritional status of the shoot (*Goebel, 1908*). According to this hypothesis, leaves produced early in shoot development are typically small and morphologically simple because leaf development is arrested by low endogenous nutrient levels, and leaves become more complex as the size and metabolic capacity of the plant increases. Subsequent studies in both ferns (*Allsopp, 1953a*, *1953b*, *1955*; *Wetmore, 1953*; *Steeves and Sussex, 1957*; *Sussex and Clutter, 1960*; *Gottlieb and Steeves, 1965*) and flowering plants (*Njoku, 1956*, *1971*; *Röbbelen, 1957*; *Feldman and Cutter, 1970*)

*For correspondence:
spoethig@sas.upenn.edu

Competing interests: The authors declare that no competing interests exist.

**eLife digest** Like animals, plants go through several stages of development before they reach maturity, and it has long been thought that some of the transitions between these stages are triggered by changes in the nutritional status of the plant. Now, based on experiments with the plant *Arabidopsis thaliana*, Yang et al. and, independently, Yu et al. have provided fresh insights into the role of sugar in 'vegetative phase change'—the transition from the juvenile form of a plant to the adult plant.

The new work takes advantage of the fact that vegetative phase change is controlled by two genes that encode microRNAs (MIRNAs). *Arabidopsis* has eight *MIR156* genes and both groups confirmed that supplying plants with sugar reduces the expression of two of these—*MIR156A* and *MIR156C*—whereas sugar deprivation increases their expression. Removing leaves also leads to upregulation of both genes, and delays the juvenile to adult transition. Given that this effect can be partially reversed by providing the plant with sugar, it is likely that sugar produced in the leaves—or one of its metabolites—is the signal that triggers the juvenile to adult transition through the reduction of miR156 levels.

Consistent with this idea, Yang and co-workers revealed that mutant plants that are deficient in chlorophyll show elevated levels of miR156 and a delayed transition to the adult form. In addition, they showed that a gene called *HXK1*, which encodes a glucose signaling protein, helps to keep plants in the juvenile form under conditions of low sugar availability. HXK1 also contributes to the glucose-induced decrease in miR156 levels and does so, at least in part, by regulating the transcription of *MIR156A* and *MIR156C* genes into messenger mRNA. HXK1 is not solely responsible for the juvenile to adult transition, however, because plants that lack this protein are only slightly precocious in their transition to the adult form.

The works of Yang et al. and Yu et al. have thus provided key insights into the mechanisms by which a leaf-derived signal controls a key developmental change in plants. Just as fruit flies use their nutritional status to regulate the onset of metamorphosis, and mammals use it to control the onset of puberty, so plants use the level of sugar in their leaves to trigger the transition from juvenile to adult forms.

have shown that exogenous sugar promotes the production of larger and more complex leaves, whereas growing plants or isolated leaf primordia in nutrient-deprived conditions or low light promotes the production of simple leaves. However, the implications of these results for the regulation of vegetative phase change remain controversial. In particular, it is still unclear if the leaf types produced by modifying carbohydrate levels correspond to juvenile and adult leaves. For example, *Jones (1995)* reported that the developmental anatomy of *Cucurbita argyrosperma* leaves produced under low light conditions is not identical to that of juvenile leaves. Defining the molecular basis for the effect of sugar on shoot morphogenesis is therefore an important problem in plant development.

Sugars regulate plant physiology and development by several different signaling pathways, which interact with various hormone and nutrient response pathways, resulting in complex signaling networks (*Rolland et al., 2006*; *Smeekens et al., 2010*). The function of HEXOKINASE1 (HXK1) in sugar signaling has been particularly well characterized (*Moore et al., 2003*; *Rolland et al., 2006*). HXK1 has dual roles in glucose homeostasis. Its enzymatic function is to catalyze the first step in glycolysis, the transformation of glucose into glucose-6-phosphate. It also functions as a sugar sensor, and can regulate gene transcription in response to changes in glucose concentration. This latter function was identified when it was discovered that glucose phosphorylation—which is a major output of the catalytic function of HXK1—was not correlated with the quantitative indicators of glucose signaling such as chlorophyll production and photosynthetic gene expression (*Moore et al., 2003*). Mutated forms of HXK1 that lack glucose phosphorylation capacity are nevertheless capable of mediating glucose signaling, demonstrating that the signaling function of HXK1 is independent of its enzymatic activity. This signaling function is executed by an HXK1-containing nuclear complex, which is thought to bind to the promoters of various genes in the presence of high levels of glucose to control their transcription (*Cho et al., 2006*).

We investigated the role of sugar in vegetative phase change by examining the effect of a photosynthetic mutation and exogenous sugar on the expression of miR156 in *Arabidopsis*. Given that miR156 regulates all aspects of vegetative phase change, we reasoned that factors that affect its

expression are likely to play an important role in this process. We show that glucose promotes vegetative phase change by repressing the accumulation of miR156 and that this effect can be attributed to a decrease in the expression of two of the eight *MIR156* genes in *Arabidopsis*. HXK1 is required for this effect, and acts through its signaling function to promote the expression of miR156 under conditions of low sugar availability. Finally, the effect of glucose on miR156 expression in defoliated seedlings supports the hypothesis that glucose, or a derivative of this sugar, is a component of the leaf signal that promotes vegetative phase change. These results provide a potential molecular mechanism for the temporal decrease in miR156 during vegetative phase change, linking nutrient levels to shoot maturation in plants.

## Results

### Reduced photosynthesis delays vegetative phase change

In *Arabidopsis*, the onset of the adult phase is characterized by the appearance of abaxial trichomes and a change of leaf shape from round leaves with a smooth margin to elongated leaves with a serrated margin (*Telfer et al., 1997*). In the original description of heteroblasty in *Arabidopsis*, *Röbbelen (1957)* reported that several chlorophyll-deficient mutations increased the number of juvenile leaves, and concluded from this observation that the production of adult leaves was promoted by the products of photosynthesis. We reinvestigated this observation by characterizing the phenotype of *chlorina1-4* (*ch1-4*), a mutation in chlorophyll *a* oxygenase (AtCAO, At1g44446) that blocks the biosynthesis of chlorophyll b (*Espineda et al., 1999*; *Oster et al., 2000*). *ch1-4* plants were yellow-green, grew more slowly than normal, and also underwent vegetative phase change significantly later than wild-type plants; under short day (SD) conditions, abaxial trichome production was delayed by five leaves and mutant leaves were rounder and had a smoother margin than the leaves of wild-type plants (*Figure 1A,B*). *ch1-4* plants did not display a delay in abaxial trichome production under long days (LD), although their leaves were still rounder than normal. To determine if the delayed phase change phenotype of *ch1-4* depends on miR156, we crossed a miR156 target site mimic construct, *35S::MIM156*, into *ch1-4* to block miR156 activity (*Wu et al., 2009*). Like *ch1-4*, plants homozygous for *ch1-4* and *35S::MIM156* were yellow-green and small in size; however, in contrast to *ch1-4*, their leaves were elongated and serrated and they produced abaxial trichomes on the first rosette leaf (*Figure 1C,D*). Thus, the delayed phase change phenotype of *ch1-4* requires miR156 activity.

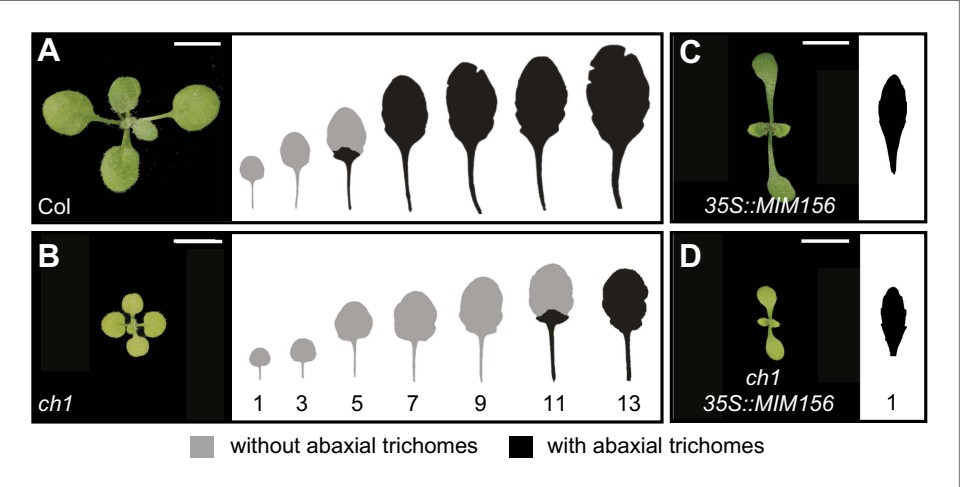

**Figure 1**. The prolonged juvenile phase of *ch1* is suppressed by *35S::MIM156*. (**A**) Wild-type Col produces about six juvenile leaves (5.7 ± 0.9, N = 24) in SD conditions. (**B**) *ch1-4* produces significantly more juvenile leaves (10.9 ± 0.8, N = 24, p<0.01) than Col. (**C**) *35S::MIM156*. (**D**) *ch1-4 35S::MIM156*. This genotype has the morphology of *35S::MIM156* and the yellow-green phenotype of *ch1-4*; double mutants produced abaxial trichomes on leaf 1 and had elongated serrated leaves, demonstrating that the delayed phase change phenotype of *ch1-4* is dependent on miR156. Numbers indicate the position from the base of the plant. Scale bar = 5 mm.

To further explore this observation, we examined the effect of *ch1-4* on the expression of miR156 and its direct targets, SPL genes. Northern blots showed that miR156 was initially more abundant and declined more slowly in *ch1-4* than in wild-type plants: miR156 expression declined to a basal level between 12 and 16 days after planting (DAP) in wild-type plants, and between 16 and 20 days DAP in *ch1-4* (*Figure 2A*). Although the *ch1-4* mutants grew more slowly than Col, the difference in miR156 accumulation is unlikely to be a consequence of delayed development because *ch1-4* mutants at 16 DAP had the same number of leaves as wild-type plants at 12 DAP, but still had more miR156 than 12-day-old wild-type plants (*Figure 2A,D*). Consistent with the elevated expression of miR156, the transcript levels of several direct targets of miR156 (*SPL3, SPL9, SPL13*) were reduced in *ch1-4* (*Figure 2B*). This effect was also reflected in the abundance of the SPL3 and SPL9 proteins, as revealed by GUS translational fusion constructs: the expression of both *pSPL3::GUS-SPL3* and *pSPL9::SPL9-GUS* was significantly lower in *ch1-4* than in wild-type plants at comparable stages of morphological development (*Figure 2C*). This effect is not attributable to an effect of *ch1-4* on the transcription of these genes because there was no significant difference in the expression of the miR156-resistant versions of these reporters in wild-type and mutant plants (*Figure 2C*). The spatial expression pattern of the miR156-sensitive version of these reporters is exemplified by *pSPL3::GUS-SPL3*, which is expressed at a very low level in cotyledons and in the first two leaves, and at a much higher level starting with leaf 3; the miR156-resistant version of this reporter, *pSPL3::GUS-rSPL3*, was expressed uniformly, and at a very high level, throughout shoot development, demonstrating that the expression pattern of *pSPL3::GUS-SPL3* is largely dependent on miR156 (*Figure 2D*). Along with the delayed phase change phenotype

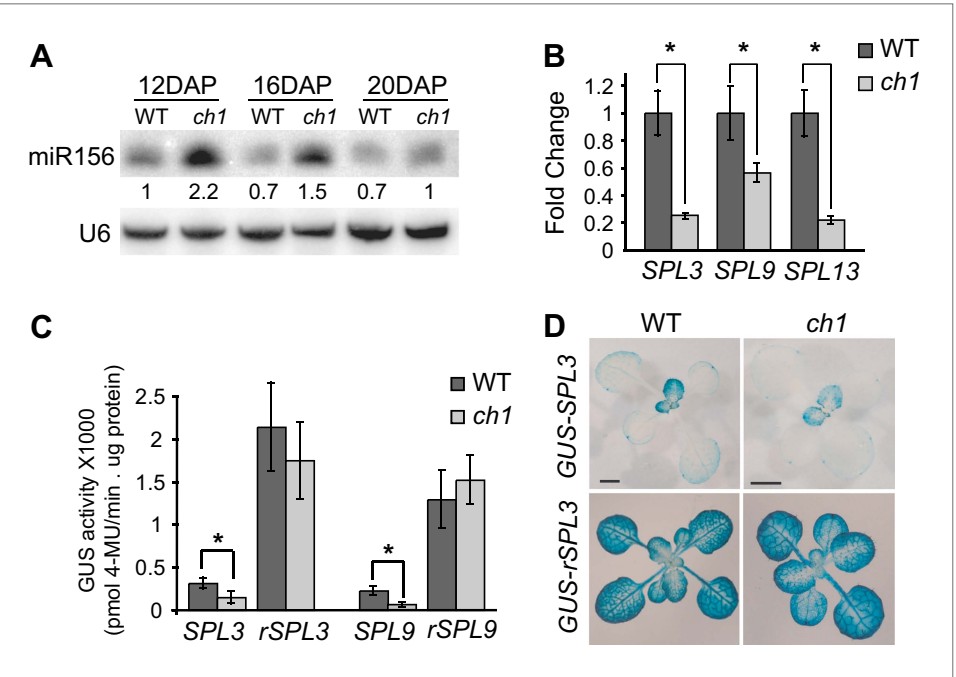

**Figure 2**. Expression of miR156 and *SPL* transcripts in *ch1-4*. (**A**) Northern blot of mature miR156 in *ch1-4* and Col reveals that miR156 is elevated in *ch1-4* and declines at a slower rate in this mutant. U6 was used as a loading control. Hybridization intensities are compared to the value in WT 12 DAP. (**B**) qRT-PCR of the transcripts of *SPL3, SPL9*, and *SPL13* in 16-day-old wild-type and *ch1-4* demonstrates that these transcripts are present at a significantly lower level in *ch1-4*. (**C**) MUG assay of the GUS activity of miR156-sensitive and miR156-resistant *SPL3 and SPL9* reporters in Col and *ch1-4* demonstrates that the reduced expression of GUS-SPL3 and SPL9-GUS in *ch1* is dependent on miR156. (**D**) The expression pattern of miR156-sensitive and miR156-resistant *pSPL3::GUS-SPL3* in 14-day-old rosettes of Col and 18-day-old rosettes of *ch1-4*; older rosettes of *ch1-4* were chosen to compensate for the slower growth rate of this mutant. *pSPL3::GUS-SPL3* is expressed at a lower level in *ch1-4*. *pSPL3::rSPL3-GUS* is expressed at the same level and in the same pattern in Col and *ch1-4*. Scale bar = 2 mm. Asterisks in (**B**) and (**C**) indicate significant difference (Student's t-test), p<0.01, n = 3. Error bars indicate SEM.

of *ch1-4*, these results suggest that the products of photosynthesis promote vegetative phase change by repressing the expression of miR156.

## Exogenous sugar represses miR156

Sugar is the major output of photosynthesis, so we explored the effect of exogenous sugar on miR156 expression and vegetative phase change. Wild-type plants were grown on plates containing 10 mM glucose or fructose, as well as several metabolically inactive sugars (mannitol, sorbitol, O-methyl-glucose). Glucose and fructose repressed the accumulation of miR156 by about 50%. Mannitol, sorbitol and O-methyl-glucose had no repressive effect on the level of miR156 (*Figure 3A*), demonstrating that the effect of glucose and fructose is not attributable to a change in osmotic pressure. miR156 levels in glucose-treated seedlings declined within 4 hr after treatment, and this effect was inhibited by cycloheximide, suggesting that it is dependent on protein synthesis (*Figure 3B*). We used *ch1-4* as a sensitized background in order to test the response of miR156 to changes in glucose levels. Mutant seedlings were grown on media containing a range of glucose concentrations. The lowest concentration, 0.5 mM glucose, produced a 50% reduction in the steady-state level of miR156, and higher concentrations produced no additional decrease (*Figure 3C*). Thus, miR156 expression is exquisitely sensitive to exogenous sugar at this early stage of development.

To determine if the sugar-induced repression of miR156 is functionally significant, and to exclude the possibility that this effect is a secondary result of a change in the overall growth of the shoot, we examined the expression of *SPL9-GUS* reporters in isolated leaf primordia. 5-mm-long primordia of leaf 6 from plants transformed with miR156-sensitive or miR156-resistant *SPL9-GUS* reporters were cultured in the presence of 10 mM of various sugars. Consistent with their effect on miR156 levels, glucose and fructose produced a significant increase in the expression of GUS-SPL9, whereas mannitol, sorbitol, and O-methyl-glucose had no effect on the expression of this transgene (*Figure 3D*). The expression of the miR156-insensitive reporter (*rSPL9-GUS*) was unaffected by glucose and fructose, indicating that the effect of these sugars on the expression of the miR156-sensitive reporter is attributable to miR156 (*Figure 3D*). The observation that SPL9-GUS activity increased after only 8 hr in culture suggests that the sugar-induced reduction in miR156 levels is unlikely to be a secondary result of a change in growth rate and—along with the rapid response

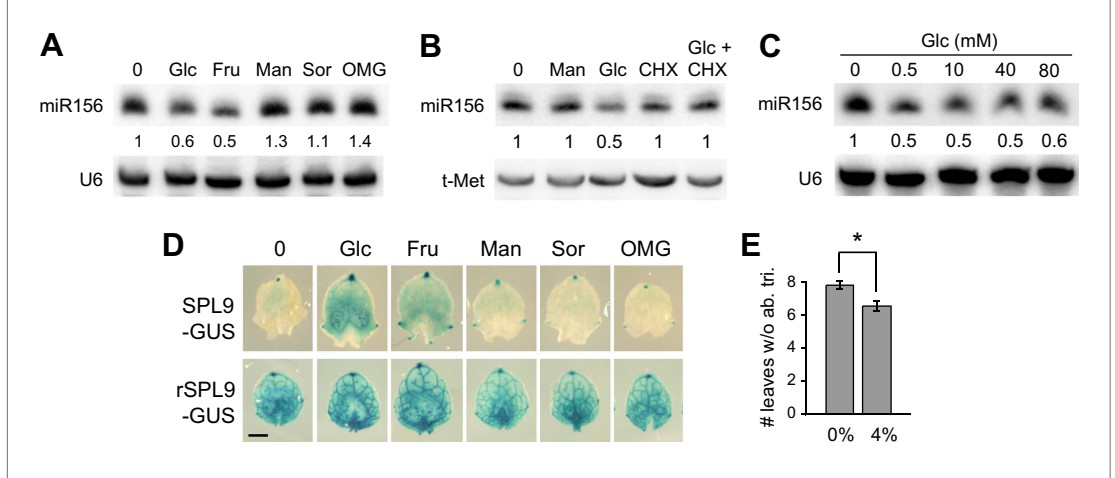

**Figure 3**. Sugar represses miR156 expression. (**A**) Northern blot of miR156 in 12-day-old plants treated with 10-mM glucose (Glc), fructose (Fru), mannitol (Man), sorbitol (Sor), and O-methyl-glucose (OMG). U6 was used as a loading control. Only Glc and Fru reduce miR156 expression. (**B**) Northern blot of miR156 in 12-day-old plants treated for 4 hr with 10 mM of the indicated substances. The effect of glucose on miR156 expression is blocked by cycloheximide, a protein synthesis inhibitor. Man: mannitol; Glc: glucose; CHX: cycloheximide. (**C**) Northern blot of miR156 in 12-day-old *ch1-4* plants treated with different amounts of glucose. 0.5 mM produced a 50% reduction in miR156, and higher amounts of glucose did not produce a further reduction. (**D**) 5-mm primordia of leaf 6 from *pSPL9::SPL9-GUS* and *pSPL9::rSPL9-GUS* plants, cultured for 8 hr in media containing 10 mM of different sugars. Scale bar = 1 mm. (**E**) Exogenous sucrose significantly accelerates abaxial trichome production in wild-type plants. Asterisk indicates significant difference (p<0.01; n = 12; error bars indicate SEM).

of miR156 to glucose (*Figure 3B,C*)—points to a direct role for sugar in the regulation of miR156 expression.

As an additional test of the functional significance of the effect of sugar on miR156 expression, we measured abaxial trichome production in wild-type plants grown on a medium containing 4% sucrose. We used sucrose for this experiment because it produces fewer deleterious effects on plant growth in this long-term experiment than glucose. Under SD conditions, sucrose significantly reduced the number of leaves without abaxial trichomes (*Figure 3E*), supporting the conclusion that the effect of sugar on miR156 and *SPL* expression is functionally important.

## Sugar represses the transcription of MIR156 genes that regulate vegetative phase change

miR156 is encoded by eight genes in *Arabidopsis*, whose specific expression patterns and individual contributions to shoot development are still unknown. Northern analysis and RNA sequencing experiments demonstrate that these genes produce 20-nt and 21-nt forms of miR156; the 20-nt form declines in abundance during shoot development, whereas the 21-nt form is expressed uniformly (*Wu and Poethig, 2006*; *Wu et al., 2009*). To determine the functional significance of the effect of sugar on miR156 expression, we characterized the expression pattern and mutant phenotypes of several of the genes that encode miR156, and examined the effect of sugar on the expression of these genes.

The expression of different *MIR156* genes in 1-mm leaf primordia was measured by qRT-PCR, using primers specific for the primary transcript (pri-miRNA) of each gene. Pri-miRNAs are processed rapidly, so it is often difficult to detect these transcripts, even by qRT-PCR. To get around this problem, we took advantage of *se-1*, a mutation in miRNA processing that causes pri-miRNAs to accumulate to relatively high levels (*Lobbes et al., 2006*; *Yang et al., 2006*; *Laubinger et al., 2008*). Pri-miRNAs derived from *MIR156A*, *MIR156B*, *MIR156C*, *MIR156D*, *MIR156F*, and *MIR156H* could be reliably detected in *se-1*, although *MIR156F* and *MIR156H* were expressed at significantly lower levels than the other four miRNAs. *MIR156E* and *MIR156G* transcripts were undetectable. An analysis of the expression of the pri-miRNAs of *MIR56A/B/C/D* revealed that only *MIR156A* and *MIR156C* are strongly down-regulated in successive 1-mm leaf primordia (*Figure 4A*). The abundance of pri-MIR156A decreased five-fold between leaves 1 and 2, and leaf 3, and was 100-fold less abundant in leaf 7 than leaves 1 and 2. The pri-MIR156C did not decrease significantly between leaves 1 and 2 and leaf 3, but declined 10-fold by leaf 7.

We determined the structure of the primary transcripts of *MIR156A* and *MIR156C* using 5′ and 3′ RACE (*Figure 4B*). This analysis was carried out using wild-type plants to avoid any potential aberrations introduced by *se-1*. We observed two major transcripts for *MIR156A*, confirming a previous study (*Xie et al., 2005*). One transcript has two exons and is 1504 nt in length; the second consists of a single 1479-nt transcript that overlaps but is slightly longer than the second exon of the first transcript (*Figure 4B* and *Supplementary file 1*). qRT-PCR performed with primers that specifically amplify these transcripts demonstrated that they are equally abundant and have the same temporal expression pattern (data not shown). The primary transcript of *MIR156C* consists of three exons and is 857-nt long (*Figure 4B* and *Supplementary file 2*).

T-DNA insertions located in the transcribed regions of these genes were identified in publically available collections. *mir156c-1* (GT22288) was originally isolated in a Landsberg *erecta* (L*er*) background, and was crossed to Col six times so that its phenotype could be compared with the *MIR156A* mutations, which were originally isolated in a Col background. *mir156a-1* (SALK_131562), *mir156a-2* (SALK_056809), and *mir156c-1* reduced the abundance of the 20-nt, but not the 21-nt form, of miR156, demonstrating that *MIR156A* and *MIR156C* specifically encode this isoform of miR156. Plants doubly mutant for *mir156c-1* and *mir156a-2* had very low levels of the 20-nt isoform of miR156, indicating that these genes are the major source of this transcript (*Figure 4C*). Plants individually mutant for *mir156a-1*, *mir156a-2*, and *mir156c-1* had no obvious morphological defects; however, plants doubly mutant for *mir156a* and *mir156c* were precocious. Abaxial trichome production started with leaf 3 (instead of leaf 5 or 6), and all of the rosette leaves of double mutants were larger, more elongated, and more highly serrated than the leaves at corresponding positions on wild-type plants (*Figure 4D,E*). Double mutants also produced four fewer rosette leaves (*Figure 4E*), flowered 1.5 days earlier (25.8 ± 0.4 vs 27.2 ± 0.4, N = 16), and had a slower rate of leaf initiation (*Figure 4F*) than wild-type plants. Along with the expression data presented

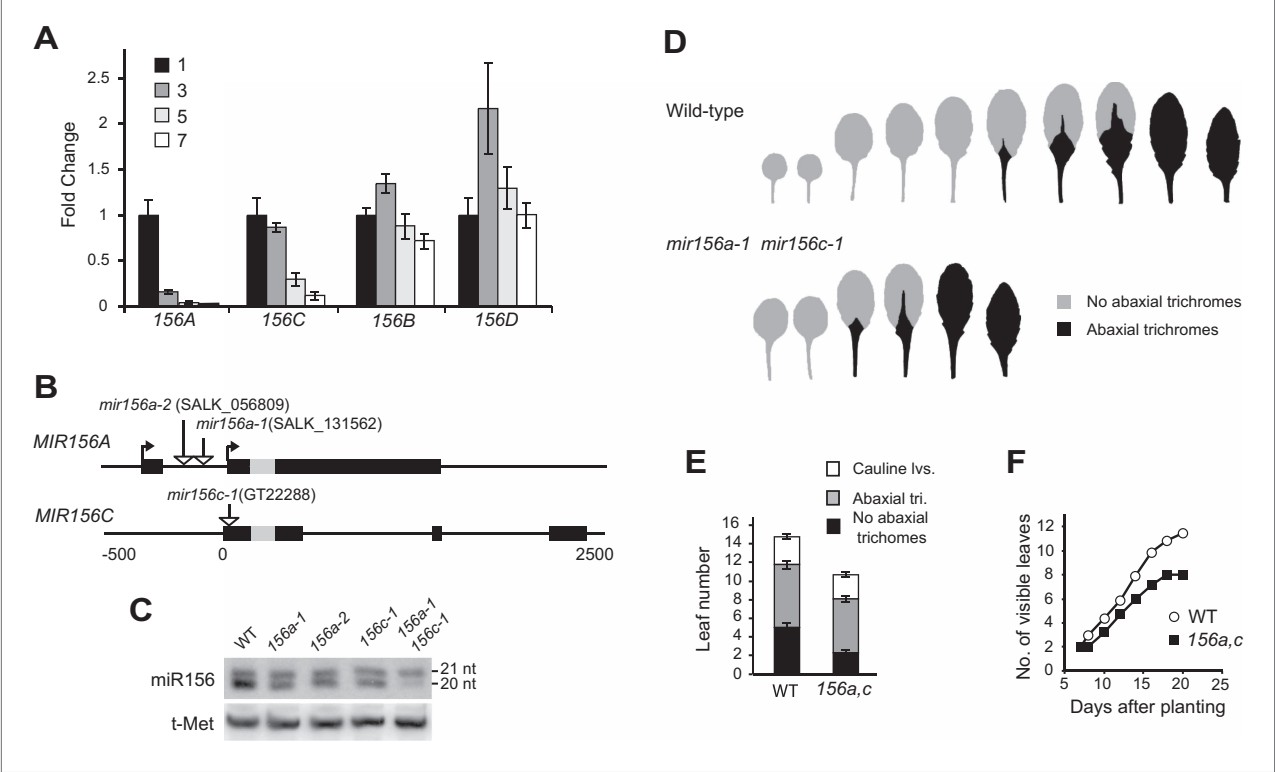

**Figure 4.** *MIR156A* and *MIR156C* are important for vegetative phase change. (**A**) qRT-PCR of primary miRNAs in 1-mm leaf primordia from different positions in *se-1* shoots, counting from the base of the rosette. Only *MIR156A* and *C* are temporally expressed. (**B**) The genomic structure of *MIR156A* and *C* and the location of T-DNA insertions in these genes. Boxes indicate exons, with the position of the miR156 hairpin indicated in gray. Arrows indicate the transcription start sites for the major *MIR156A* transcripts. (**C**) Northern blot of miR156 in single and double mutants of *MIR156A* and *MIR156C*. These mutations only affect the accumulation of the 20-nt form of miR156, demonstrating that the 21-nt form is the product of another gene or genes. (**D**) Rosette leaves of wild-type and *mir156a-1 mir156c-1* plants grown in LD. (**E**) The *mir156a-1 mir156c-1* double mutant has significantly fewer juvenile leaves and transition/adult leaves (error bars indicate SEM). (**F**) The *mir156a-1 mir156c-1* double mutant has a reduced rate of leaf initiation.

above, this loss-of-function phenotype demonstrates that *MIR156A* and *MIR156C* are key regulators of vegetative phase change.

Glucose reduced the abundance of the pri-miRNAs of *MIR156A*, *C*, *F*, and *H*, but had no effect on *MIR156B* and *D* (**Figure 5A**). This result, and the observation that *MIR156A* and *MIR156C* are expressed at much higher levels than *MIR156F* and *MIR156H*, suggests that glucose affects level of miR156 primarily through its effect on *MIR156A* and *MIR156C*. To study the mechanism by which sugar regulates the expression of these genes, we produced transgenic plants containing genomic constructs in which the *MIR156A* or *MIR156C* hairpin sequences were replaced with GUS+ (**Figure 5B**). These 7-kb constructs contained the entire intergenic regions upstream and downstream of *MIR156A* and *MIR156C*, and were highly expressed in young seedlings. Both glucose and sucrose reduced the abundance of the endogenous pri-MIR156A and pri-MIR156C transcripts in these reporter lines to about 20% of the wild-type level. However, glucose had little or no effect on the expression of the MIR156A-GUS and MIR156C-GUS reporters. In the experiment shown in **Figure 5B and C**, 50 mM glucose had no apparent effect on MIR156A-GUS expression and reduced MIR156C-GUS expression to approximately 70% of the wild-type level, but in other experiments, 50 mM glucose had no effect on the expression of either reporter. 50 mM sucrose reduced the expression of both reporters to approximately 50% of wild type (**Figure 5C**), and seedlings grown in the presence of 120 mM sucrose displayed an even more substantial reduction in expression (**Figure 5—figure supplement 1**). These results suggest that sucrose regulates *MIR156A* and *MIR156C* expression at a transcriptional level, but raise the possibility that glucose regulates one or both of these genes by a different mechanism.

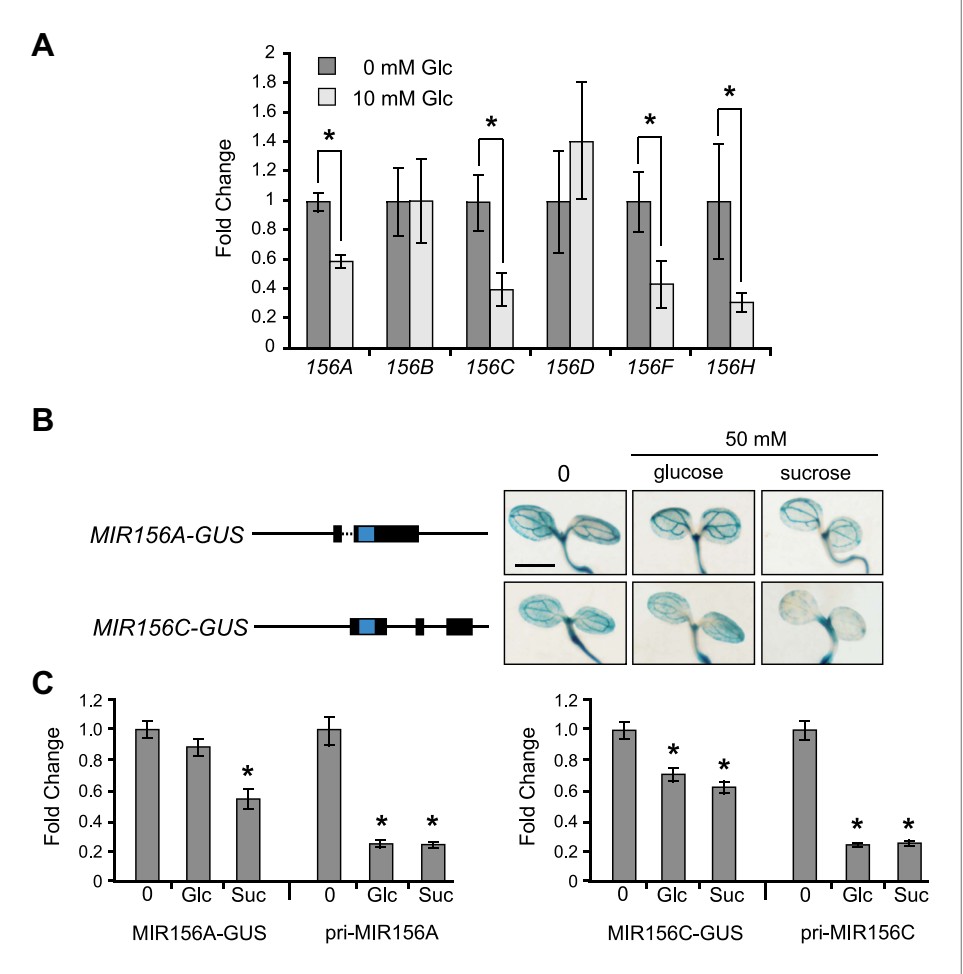

**Figure 5**. Sugar specifically regulates the expression of *MIR156* genes that are important for vegetative phase change. (**A**) qRT-PCR of primary MIRNAs in 12-day-old *se-1* plants grown in the absence or presence of 10 mM glucose. *MIR156A, C, F*, and *H* are repressed by glucose. Asterisk indicates significantly different from the no sugar treatment, $p < 0.01$. (**B**) The structure of *MIR156A-GUS* and *MIR156C-GUS* reporter constructs, and their response to 50 mM glucose or sucrose. Blue indicates the location of GUS, which was inserted in place of the miR156 hairpin. The full-length constructs are approximately 7 kb in length. Solid line: intergenic sequence; dashed line: intron; bar = 2 mm. The staining response was representative of two independent lines, homozygous for a single transgenic insertion. (**C**) qRT-PCR analysis of the abundance of the MIR156A-GUS, MIR156C-GUS, pri-MIR156A, and pri-MIR156C transcripts in the seedlings illustrated in (**B**). GUS-fusion transcripts were measured using primers specific for GUS+. Asterisk indicates significantly different from the no sugar treatment, $p < 0.01$.

The following figure supplements are available for figure 5:

**Figure supplement 1**. Sucrose represses the expression of the *MIR156A-GUS* and *MIR156C-GUS* reporter genes in transgenic seedlings.

## HXK1 is required for sugar-mediated repression of miR156 early in shoot development

Previous work has shown that HXK1 has both enzymatic and regulatory functions in glucose metabolism/signaling (*Moore et al., 2003*; *Cho et al., 2006*). We explored the possibility that *HXK1* mediates the effect of sugar on vegetative phase change by examining the phenotype of a null allele of *HXK1*, *gin2-1*, which was isolated in a L*er* background. Northern analysis revealed that 8-day-old *gin2-1* seedlings growing in soil had less miR156 than wild-type plants, whereas 11-day-old seedlings had the same low level of miR156 as wild-type plants (*Figure 6A*). Thus, HXK1 contributes to the elevated level of

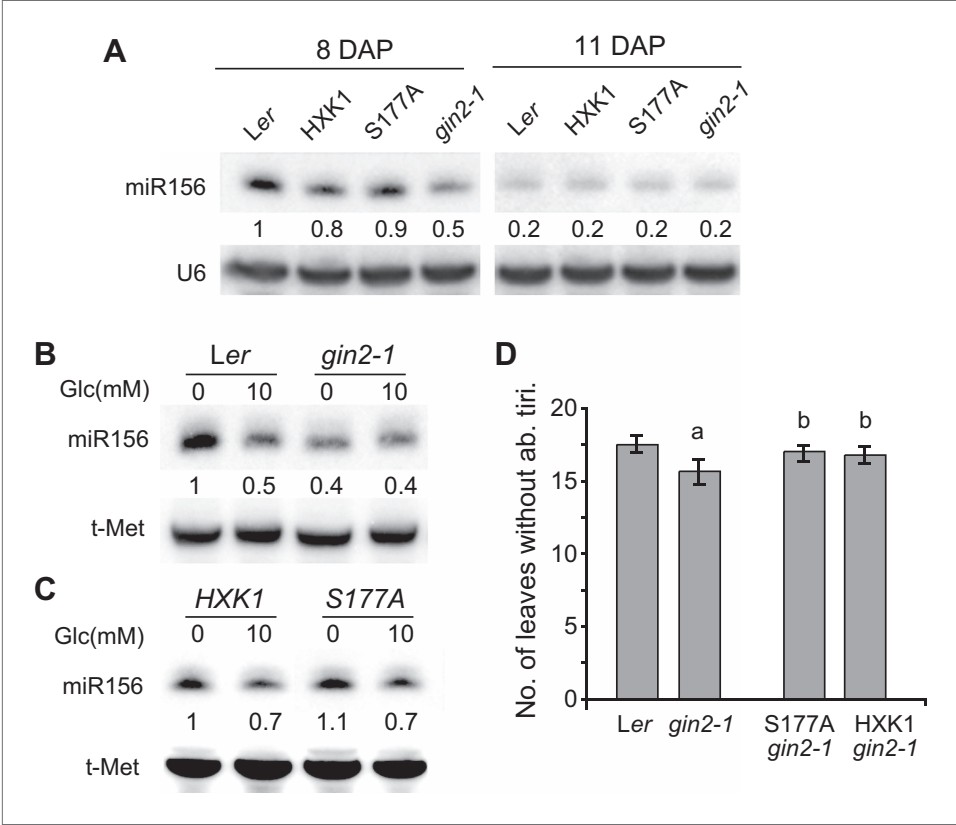

**Figure 6**. The signaling role of HXK1 is required for glucose-mediated repression of miR156. (**A**) Northern blot of miR156 in 8- and 11-day-old L*er* and *gin2-1* seedlings grown in soil. *gin2-1* has less miR156 than wild-type seedlings 8 days after planting, but has normal levels of miR156 at later stages. Hybridization intensities are compared to the intensity in L*er*, 8 DAP. (**B**) Northern blot of miR156 in L*er* and *gin2-1* grown in the absence and presence of glucose. *gin2-1* has lower levels of miR156 than L*er* in the absence of glucose and is not further repressed by exogenous glucose. (**C**) Northern blot of miR156 in *gin2-1* plants transformed with a wild-type *HXK1* construct and a construct (*S177A*) that lacks enzymatic activity. miR156 accumulation was repressed by glucose in these transgenics by approximately the same amount as in L*er*. (**D**) The number of leaves without abaxial trichomes in L*er*, *gin2-1*, *HXK1/gin2-1*, and *S177A/gin2-1* grown under SD, 16°C, and a light intensity of 60 μmol/m²/s. *gin2-1* has significantly fewer juvenile leaves than L*er* and the transgenic lines (a, n = 48, p<0.01, error bars indicate SEM). The number of juvenile leaves in *gin2-1* plants transformed with wild-type (*HXK1*) and enzymatically inactive (*S177A*) HXK1 was not significantly different from wild-type L*er* (b, n = 48, p>0.1, error bars indicate SEM).

The following figure supplements are available for figure 6:

**Figure supplement 1**. HXK1 promotes the accumulation of miR156 in the absence of exogenous sugar.

**Figure supplement 2**. The timing of abaxial trichome production in wild-type *gin2-1* and *gin2-1* plants transformed with a wild-type (HXK1) or catalytically inactive form (S177A) of HXK1.

miR156 in juvenile plants, but plays a minor role in the expression of miR156 at later stages of shoot development. To determine if HXK1 is required for the response to exogenous sugar, we examined miR156 levels in 10-day-old L*er* and *gin2-1* seedlings grown in the presence and absence of 10 mM glucose (these seedlings were at the same developmental stage as 8-day-old soil-grown seedlings). Glucose reduced the abundance of miR156 in L*er*, but had no effect on miR156 levels in *gin2-1* (***Figure 6B*** and ***Figure 6—figure supplement 1A***). miR156 levels were also examined in *gin2-1* plants transformed with a catalytically inactive form of HXK1 (*S177A/gin2-1*) (***Moore et al., 2003***; ***Cho et al., 2006***) to determine if the effect of HXK1 is attributable to its enzymatic or signaling function. *gin2-1* plants expressing the catalytically inactive protein (*S177A/gin2-1*) or the wild-type HXK1 protein (*HXK1/gin2-1*)

displayed the same pattern of miR156 expression as L*er* in sugar-deficient and sugar-supplemented media (*Figure 6A,C* and *Figure 6—figure supplement 1*). Thus, the enzymatic function of HXK1 is not responsible for promoting miR156 expression in sugar-deficient conditions. *gin2-1* produced a small, but statistically significant, decrease in the number of leaves without abaxial trichomes, and this phenotype was completely rescued by both the *S177A* and the wild-type *HXK1* transgenes (*Figure 6D*). L*er* has very short juvenile phase (3–4 leaves) in our normal SD growing conditions (22°C, 10 hr light at 220 µmol/m²/s), making it difficult to detect the effect of mutations that accelerate vegetative phase change; the effect of *gin2-1* was quite small under these conditions, albeit statistically significant (*Figure 6— figure supplement 2*). Growing plants in SD at 16°C under relatively low light (60 µmol/m²/s) lengthened the juvenile phase, and increased the difference between wild-type and *gin2-1* plants (*Figure 6D*). This precocious phenotype is consistent with the reduced level of miR156 in *gin2-1*, and suggests that this decrease in miR156 is functionally significant. In summary, these results indicate that HXK1 promotes the accumulation of miR156 early in shoot development under conditions of low sugar availability, and contributes to the sugar-mediated repression of this miRNA. However, it is clearly not the only factor involved in the developmental decline in miR156 because this decline occurs even in *gin2-1* (*Figure 6A*).

## Glucose suppresses the defoliation-induced increase of miR156

We previously reported that vegetative phase change is promoted by a leaf-derived signal that represses the transcription of *MIR156A* and *MIR156C*, and this mechanism is conserved in other plant species (*Yang et al., 2011*). To test if sugar could serve as this leaf signal, we took advantage of *Nicotiana benthamiana*, which has significantly larger seedlings than *Arabidopsis* and, like *Arabidopsis*, produces elevated levels of miR156 in response to defoliation (*Figure 7A*) (*Yang et al., 2011*). We reasoned that the substance(s) that promote vegetative phase change would reverse this effect. To mimic the normal situation as much as possible, we applied candidate substances to petiole stubs immediately after defoliation, rather than testing their effect on isolated leaf primordia. We tested if this approach was experimentally feasible by applying crude extracts from 1-cm leaf primordia. Remarkably, these extracts partially suppressed the effect of defoliation on miR156 expression (*Figure 7B*). We then tested the effect of glucose, and found that 300 mM glucose was as effective as leaf extracts in reducing miR156 expression in defoliated seedlings (*Figure 7B*). This result is consistent with the hypothesis that glucose, or a metabolite derived from glucose, is the leaf factor that promotes vegetative phase change.

## Discussion

Nutrients play important roles in many developmental transitions in both animals and plants. In *Caenorhabditis elegans*, nutrients trigger the first step in the heterochronic pathway that regulates larval maturation, whereas the dauer developmental arrest pathway is initiated by nutrient-deficient conditions (*Rougvie, 2005*). Nutrients regulate the timing of metamorphosis in *Drosophila* through their effect on the TOR nutrient-sensing pathway (*Layalle et al., 2008*), and in mammals, nutrient levels affect the onset of puberty (*Tolson and Chappell, 2012*). In plants, evidence that nutrients play an important role in shoot maturation was obtained over a century ago (*Goebel, 1908*), and led to many subsequent studies of the effect of carbohydrates and other nutrients on leaf and shoot development (*Ashby, 1948*; *Allsopp, 1967*). Our work extends these classical studies by demonstrating that sugar regulates shoot maturation in *Arabidopsis* by repressing the expression of a key regulator of juvenile identity, miR156. Additionally, we found that *MIR156* genes are differentially regulated and that sugar only affects a subset of these genes. Analyses of the expression patterns and mutant phenotypes of two of these sugar-repressed genes, *MIR156A* and *MIR156C*, demonstrated that they play a

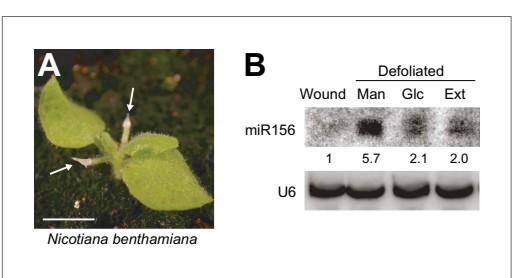

**Figure 7**. Glucose rescues the defoliation-induced increase in miR156. (**A**) 3-week-old *N. benthamiana* plant with agarose gel on petiole stubs. Scale bar = 1 cm. (**B**) Northern analysis of the level of miR156 in the shoot apex of wounded nondefoliated plants (used as a control), and defoliated plants treated with leaf extract (Ext), or 300 mM of mannitol (Man) or glucose (Glc). miR156 was assayed 3 days after treatment.

major role in vegetative phase change. These results, as well as the evidence that sugars increase the expression of the direct targets of miR156, support the hypothesis that sugars promote vegetative phase change through their effect on miR156, and suggest potential mechanisms for the temporal regulation of vegetative phase change in flowering plants.

Our results also indicate that HXK1 contributes to the sugar-mediated repression of miR156 by promoting the accumulation of miR156 under low sugar conditions. Hexokinases function as glucose sensors in both yeast and plants (*Rolland et al., 2002*), so the requirement for HXK1 suggests that glucose is at least partially involved in the sugar response. HXK1 participates in a nuclear complex to directly regulate gene expression (*Cho et al., 2006*). One possibility is that HXK1 directly promotes the transcription of *MIR156* genes. In this scenario, binding of glucose to HXK1 would block its activity, resulting in a reduction in *MIR156A* and *MIR156C* transcription. Because HXK1 does not seem to have transcriptional activity on its own, this would require an interaction between HXK1 and DNA-binding transcription factors, as has been previously described (*Cho et al., 2006*). It is also possible that sugar represses the accumulation of mature miR156 post-transcriptionally, in addition to its role in the transcriptional regulation of *MIR156A* and *MIR156C*. In this scenario, sugar operates by one pathway to repress the transcription of *MIR156A* and *MIR156C* and by a different pathway to post-transcriptionally reduce the abundance of miR156. The differential effect of glucose and sucrose on the expression of *MIR156A-GUS* and *MIR156C-GUS* may be significant in this regard. Whereas sucrose reliably decreased the expression of these reporters, they were largely insensitive to glucose. This result does not reflect a difference in the inherent sensitivity of seedlings to these sugars because glucose and sucrose were equally effective in reducing the abundance of the endogenous pri-MIR156A and pri-MIR156C transcripts. This result could reflect the absence of glucose-responding *cis*-regulatory sequences in these constructs, but this seems unlikely because we included the entire intragenic regions upstream and downstream of the miR156 hairpin. A third possibility (based on the observation that glucose decreases the abundance of the endogenous pri-MIR156A and pri-MIR156C transcripts but not transcripts in which the miR156 stem-loop is replaced by GUS) is that glucose regulates the abundance of pri-MIR156A and pri-MIR156C post-transcriptionally, by a mechanism that is dependent on the presence of the miR156 stem-loop structure. If so, glucose must act by destabilizing these primary transcripts, not by blocking the processing of the miR156 stem-loop, because a defect in miR156 processing would be expected to produce an increase in the level of pri-MIR156, not the decrease that we observed. Whatever the case may be, the relatively mild phenotype of *gin2-1* and the observation that miR156 decreases over time in *gin2-1* demonstrates that HXK1 is only one of several signaling molecules (*Smeekens et al., 2010*) involved in the regulation of miR156.

It remains to be determined if sugar levels increase in the shoot apex during vegetative phase change. Nevertheless, it is worth considering how such an increase might occur. One possibility is a regulated increase in sugar export, as occurs upon photoperiodic induction of flowering (*Lejeune et al., 1991*, *1993*; *Perilleux and Bernier, 1997*; *Corbesier et al., 2002*). However, photoperiod does not play a major role in vegetative phase change in *Arabidopsis* (*Willmann and Poethig, 2011*), so if the pattern of sugar export changes during vegetative phase change, this event must be regulated by a different factor. An alternative possibility is that vegetative phase change is simply a consequence of an increase in the overall photosynthetic output of the shoot resulting, perhaps, from an increase in leaf number, or a change in photosynthetic efficiency. The observation that the rate of photosynthesis increases significantly during vegetative phase change in *Eucalyptus* (*Cameron, 1970*), maize (*Thiagarajah et al., 1981*), and rice (*Asai et al., 2002*) is consistent with this latter hypothesis. A corollary of this hypothesis is that the juvenile phase is specifically adapted for growth under light-limited conditions, and studies of *Acacia* (*Brodribb and Hill, 1993*; *Pasquet-Kok et al., 2010*), *Eucalyptus* (*Cameron, 1970*; *Ashton and Turner, 1979*; *James and Bell, 2000*), and several other species (*Day, 1998*) indicate that this is the case.

Endogenous sugar levels fluctuate throughout the day (*Blasing et al., 2005*) and in response to various stresses (*Hummel et al., 2010*). Presumably, this fluctuation in sugar levels affects miR156 expression; yet, most plants do not revert to the juvenile phase once they are in the adult phase, and if they do, this event usually involves a few leaves or branches in a localized region of the shoot (*Schaffalitzky de Muckadell, 1954*; *Brink, 1962*), which is not the pattern expected from a global change in nutrient levels. What accounts for the stability of these phases? One possible explanation is that the sensitivity to sugar in the adult phase is overcome by stable epigenetic modification of *MIR156* loci. An alternative possibility is that miR156 expression is only strongly affected by sugar below a threshold concentration of sugar. If sugar levels are above this threshold—as they are expected to be

in the adult phase—a further increase might have minor effects on miR156 expression, but this variation would not be sufficient to affect shoot identity. In this regard, it is significant that *gin2-1* had lower levels of miR156 than wild-type plants in the absence of exogenous glucose, but did not differ from wild type in the presence of glucose. The implication of this observation is that HXK1 only promotes miR156 expression under low sugar conditions; above a threshold concentration, glucose blocks HXK1 activity, resulting in a decrease in miR156 expression and buffering miR156 expression against fluctuations in glucose concentration. This hypothesis is supported by the observation that in *ch1-4* miR156 is only elevated in the juvenile phase; adult phase *ch1-4* plants had the same level of miR156 as wild-type plants, although there was no obvious improvement in their growth rate or color that might indicate an increase in their photosynthetic capacity.

The results presented here provide molecular evidence for the long-standing hypothesis that states that maturation is regulated by an increase in endogenous carbohydrates (**Allsopp, 1954**). Additional studies will be required to determine if carbohydrates are the primary regulators of vegetative phase change, or if other endogenous factors play an important role in this developmental transition.

## Materials and methods

### Genetic stocks

Unless otherwise specified, all stocks were in a Col genetic background. *ch1-4* was a gift from Roberto Bassi (University of Verona), *mir156a-1* (SALK_056809) and *mir156a-2* (SALK_131562) were obtained from the Arabidopsis Biological Resource Center (Columbus, OH, United States), and *mir156c-1* (GT22288) was a gift from Robert Martienssen. *mir156c-1* was originally in a L*er* background, and was crossed to Col six times, accompanied by PCR-genotyping to minimize linkage drag. The stock used for this study contains an L*er* segment of less than 1 Mb surrounding *mir156c-1*, but is otherwise Col. *gin2-1*, *HXK1/gin2-1*, and *S177A/gin2-1* seeds were obtained from Brandon Moore (Clemson University) and are in L*er*.

### Growth conditions

Seeds were sown on fertilized Fafard #2 soil (Fafard) and left at 4°C for 2 days prior to being transferred to a growth chamber. Plant age (days after planting, DAP) was measured from the time the seeds were transferred to the growth chamber. Plants were illuminated with a 3:1 combination of cool white (F032/841/Eco; Sylvania) and wide spectrum (Gro Lite WS; Interlectric Corp.) fluorescent lights under LD (16 hr light:8 hr dark; 140 µmol/m$^2$/s; 22°C) or SD conditions (10 hr light:14 hr dark, 200 µmol/m$^2$/s; 22°C). Unless otherwise stated, the effect of various sugars on miR156 expression was examined in 12-day-old seedlings grow on sugar-supplemented ½ strength Murashige and Skoog medium under SD conditions.

### Small RNA blots

RNA blots were processed as described previously (**Wu and Poethig, 2006**). Briefly, plant tissues were homogenized in liquid nitrogen, and total RNAs from these tissues were extracted using TRIzol reagent (Invitrogen). To isolate small RNAs, the total RNAs were incubated on ice with 500 mM NaCl and 5% PEG8000 for 2 hr and centrifuged at 13,000 rpm for 10 min. Supernatants were collected and incubated with 1/10 volume 3M NaOAc and 2 volume of 95% ethanol at −20°C for 2 hr. Small RNAs were then precipitated by centrifuging at 13,000 rpm for 10 min. The concentration of small RNAs was quantified using a nanodrop spectrophotometer before being loaded on polyacrylamide gel.

### Quantitative real-time PCR

Total RNAs extracted using TRIzol reagent (Invitrogen) were purified using a Qiagen RNeasy mini kit (Qiagen). Purified RNAs were quantified and reverse transcribed into cDNAs using Invitrogen SuperScript II Reverse Transcriptase (Invitrogen). cDNAs were diluted and used as templates for real-time PCR. PCR reactions were performed with a Power SYBR Green PCR Master Mix (Applied Biosystems) using *TUBLIN* or *UBQ10* as a standard.

### GUS staining and MUG assay

The sugar response was tested using 5-mm-long primordia of leaf 6. Leaf primordia were detached from SPL9-GUS+ and rSPL9-GUS+ reporter lines, shaken for 10 min in different media, and then kept at room temperature for 8 hr. After incubation, the leaves were submerged in X-Gluc solution, evacuated,

and kept at 37°C for 6 hr. Leaves were decolorized in 70% ethanol. The MUG assay was performed on 12-day-old shoots, as previously described (*Wu and Poethig, 2006*).

### MIR156 reporter constructs

To generate the full-length *MIR156A* reporter line, a 3-kb fragment upstream and 4-kb fragment downstream of the *MIR156A* stem-loop structure were cloned into a pCAMBIA3301-GUS+ vector at the Nco I/EcoR I and Pml I/BstE II sites, respectively. The truncated version of the *MIR156A* reporter was generated by replacing the upstream sequence with a truncated PCR product. The *MIR156C* reporter line was generated in a similar fashion, using genomic fragments 3.7 kb upstream and 3 kb downstream of the MIR156C stem-loop. More than 40 individual transgenic lines from each construct were analyzed, and transgenic lines harboring stable single insertions were selected for further analysis. Seedlings were treated with 90% acetone on ice for 10 min, and then washed with water three times before staining with X-Gluc.

### miR156a RACE

Total RNA from SD-grown 7-day-old Col seedlings was isolated as described above, then digested with RNase-free DNase1 (Qiagen), and cleaned using RNeasy columns (Qiagen). Purified RNA was dephosphorylated, decapped, ligated to RNA adaptors, and reverse transcribed according to the manufacturer's instructions (Invitrogen). The 5′ and 3′ RACE were performed with GeneRacer primers (supplied with the kit) and miR156a-specific primers: 5′-CTCTTGTCCCAACTCTTTCATTCACAATTA-3′; 5′-GTGCTGATCTCTTTGGCCTGTCTT-3′ (*Supplementary file 3*).

### Leaf ablation

The first two leaves of *Nicotiana benthamiana* plants grown in LD were removed when plants were 2 weeks old. Leaf extracts were prepared from 1-cm *N. benthamiana* leaf primordia. Leaf primordia were ground in liquid nitrogen, and mixed with 1% low melting temperature agarose at about 50°C. No centrifugation or filtration was performed before applying the extract to leaf stubs. Glucose or mannitol solutions were similarly mixed with low melting temperature agarose and then applied to leaf stubs immediately after defoliation. Shoot apices with leaves less than 1 cm long were harvested 3 days after manipulation, and miR156 levels were determined by Northern analysis, as described above.

## Acknowledgements

We are grateful for helpful discussions with members of the Poethig laboratory, and to Mark Goulian for his help with the MUG assay. We also wish to thank Keith Earley for generating the MIR156A and MIR156C reporter lines, and Sen Sen Liu and Tamra Fisher for producing preliminary data for this article.

## Additional information

### Funding

| Funder | Grant reference number | Author |
| --- | --- | --- |
| National Institutes of Health | NIH-R01-GM051893 | R Scott Poethig |

The funder had no role in study design, data collection and interpretation, or the decision to submit the work for publication.

### Author contributions

LY, RSP, Conception and design, Acquisition of data, Analysis and interpretation of data, Drafting or revising the article; MX, YK, JH, Acquisition of data, Analysis and interpretation of data

## Additional files

### Supplementary files

• Supplementary file 1. The genomic organization of *MIR156A* (At2g25095). The two most abundant transcripts are indicated. Exons are indicated in yellow. The miR156 hairpin is underlined, and the mature miRNA is indicated in blue.

• Supplementary file 2. The genomic organization of *MIR156C* (At4g31877). Exons are indicated in yellow. The miR156 hairpin is underlined, and the mature miRNA is indicated in blue. A variant 5' end of the transcript is indicated in green.

• Supplementary file 3. PCR primers and oligonucleotide probes.

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
