## [Decision Letter]

Thank you for choosing to send your work entitled “Sugar promotes vegetative phase change in *Arabidopsis thaliana* by repressing the transcription of *MIR156A* and *MIR156C*” for consideration at *eLife*. Your article has been evaluated by a Senior editor (Detlef Weigel) and 2 reviewers, one of whom (Rick Amasino) is a member of our Board of Reviewing Editors.

The Reviewing editor and the other reviewer discussed their comments before we reached this decision, and the Reviewing editor has assembled the following comments based on the reviewers' reports.

The finding that sugar may be the age signal that acts via the miR156/SPL module is a significant advance in plant biology that will stimulate further research.

There are, however, some issues that need to be resolved, in particular the inconsistencies between your paper and the related paper submitted by the Wang lab.

[Editors' note: these two studies were conducted independently; each was reviewed on its own merits, on the understanding that it would be published alongside the related study if both were accepted.]

Although the major conclusion of both papers is the same, namely that sugar treatment reduces the levels of miR156, three major differences are: 1) effect or lack thereof of cycloheximide, 2) hexokinase dependence (i.e., whether or not HXK1 is a glucose sensor for the age effect), and 3) transcriptional versus post-transcriptional regulation of *MIR156* expression. Some of these inconsistencies might result from studying different promoters (*MIR156A* vs *MIR156C*), measuring miR156 versus *pri-MIR156C* levels, the use of liquid culture versus solid media, or different temperature and light regimens.

Two other issues are:

4) The possible difference in effect of the *cao/chlorina1* mutant is due to short day versus long day conditions. In addition to day length, light intensity, and temperature are likely to play a role. (There also needs to be agreement on the nomenclature of the mutant.)

5) Differences in glucose concentrations used.

We hope that you will exchange information with Jia-Wei Wang and colleagues on experimental design, repeat a number of crucial experiments under common conditions (which should not take long), and determine if the inconsistencies may in fact be resolvable. We will relay a corresponding message to the Wang lab.

A final note: the editors and reviewers discussed the lack of measurement of endogenous sugars during phase change and concluded that, although this would be a valuable addition to the body of work you have presented, it would be beyond the scope of your study.

Comments specific to your paper:

A) Different sugar concentrations were used for the gene expression and promoter experiments; for the former 0.5–20 mM glucose is used, while 1% (∼50 mM) was used for the latter. Is there a reason for this? If so, please make that clear to the reader.

B) You should attempt to quantify GUS activity for the MIR156 promoter response experiments.

C) Regarding the signaling role of HXK1 presented in Figure 6A and 6B, how many biological replicates were performed?

D) The plant growth conditions for the *gin2* experiments are very different from those of the other experiments: plants were grown at 16ºC instead of 22ºC and the light intensity was much lower (60 vs 200 micromol). What is the rationale for this, especially since it is has been reported that *MIR156A* becomes up-regulated at lower temperatures (see e.g., Lee et al., 2010) and the differences between Ler (WT) and *gin2-1* are only marginal (Figure 6C)?

E) Furthermore, are the complemented *gin2* mutants significantly different from the *gin2* single mutants (column 2 vs 4)? If not, this affects the ability to draw conclusions about the role of HXK1 in the regulation of miR156 levels.

F) In Figure 6, it would be useful to include results for Ler + glucose when grown under the same conditions. In Figure 6A it is shown that *gin2-1* w/o glucose has similar low levels of miR156 as Ler + glucose, but it would be useful to know whether at the morphological level (# of leaves w/o abaxial trichomes) the outcome is the same.

---

## [Author Response]

*Although the major conclusion of both papers is the same, namely that sugar treatment reduces the levels of miR156, three major differences are: 1) effect or lack thereof of cycloheximide, 2) hexokinase dependence (i.e., whether or not HXK1 is a glucose sensor for the age effect), and 3) transcriptional versus post-transcriptional regulation of MIR156 expression. Some of these inconsistencies might result from studying different promoters (MIR156A vs MIR156C), measuring miR156 versus pri-MIR156C levels, the use of liquid culture versus solid media, or different temperature and light regimens*.

1) We examined the effect of cycloheximide on the level of mature miR156 and found that sugar-mediated repression of this transcript does not occur in the presence of cycloheximide. Jia-Wei's result for *MIR156A* matches our results for mature miR156. One possible explanation is that we examined the effect of cycloheximide on mature miR156 levels and Jia-Wei examined the effect on pri-miR156 transcripts. If sugar represses the accumulation of miR156 post-transcriptionally through a translation-dependent process, this may explain the difference. It also could be the case that *MIR156A* is the major contributor to the mature miR156 pool at this stage.

2) Jia-Wei reported that *gin2* mutations do not block the repressive effect of sugar on miR156 expression; in contrast, we found that the accumulation of the mature miR156 in *gin2-1* is insensitive to sugar. We tested the effect of sugar on the primary *MIR156A* and *MIR156C* transcripts in Ler and *gin2-1* by qRT-PCR, and found that *gin2-1* completely blocks the glucose-induced decrease in *pri-MIR156A* and partially blocks the decrease in *pri-MIR156C* (data not shown). These results are consistent with our northern results for the mature miR156 transcript.

It should be emphasized that our claim that HXK1 promotes miR156 expression is not based on the lack of an effect of sugar on miR156 expression in *gin2-1*, but rather on the effect of *gin2-1* on miR156 expression in the absence of exogenous sugar. We observed that *gin2-1* has lower levels of miR156 than wild-type plants on sugar-free media (i.e., in soil), leading to the conclusion that HXK1 promotes the accumulation of miR156 under conditions of low sugar.

We should also emphasize that we are not claiming that HXK1 is the only sugar sensor involved in miR156 expression, and do not believe that this is the case. In the revised manuscript, we present a new panel (Figure 5A) demonstrating that miR156 declines in *gin2-1* over time, just as it does in wild-type plants. Thus, HXK is not required for the temporal decline in miR156, but it does contribute to the initial high level of miR156 expression.

We assayed the effect of sugar on *gin2-1* using two methods: 1) directly growing seedlings on ½ MS medium containing various sugars; 2) growing seedlings on ½ MS medium without any sugar, and then transferring these seedlings to media containing various sugars for 4–12 hrs. Both methods produced similar results.

It is important to note that the effect of *gin2* is dependent on both genetic background and mineral nutrients. In contrast to *gin2-1,* which is in a Ler background, we did not observe a stable low level of miR156 in T-DNA alleles of HXK1 in a Col background. Second, we found that nitrogen in the form of nitrate or ammonia is essential for the sugar response. When we performed sugar response experiments in a nitrogen-deficient medium, the sugar-mediated regulation of miR156 disappeared. It is well known that plants are acutely sensitive to the C:N ratio, so this result is not surprising. For example, the repressive role glucose on photosynthetic gene expression is most evident in a nitrate-deficient medium (Moore et al, 2003).

3) We re-examined the effect glucose and sucrose on the *MIR156A*-GUS reporter, and expanded this analysis to *MIR156C*, using a newly-constructed *MIR156C*-GUS reporter. We confirmed that sucrose decreased the expression of the *MIR156A*-GUS reporter, and found that it suppressed the *MIR156C*-GUS reporter as well (Figure 5). Glucose had no effect on the expression of *MIR156A*-GUS and only a modest effect on *MIR156C*-GUS, even at a concentration that decreased in the expression of the endogenous *MIR156A* and *MIR156C* primary transcripts. This result suggests that sucrose represses the transcription of *MIR156A* and *MIR156C*, whereas glucose destabilizes the primary transcripts of these genes by a mechanism that might be dependent on the presence of the miR156 hairpin. These new data may explain some of the initial discrepancies between our conclusions and those of Jia-Wei.

*Two other issues are*:

*4) The possible difference in effect of the cao/chlorina1 mutant is due to short day versus long day conditions. In addition to day length, light intensity, and temperature are likely to play a role. (There also needs to be agreement on the nomenclature of the mutant.*)

The effect of *ch1* on leaf shape is independent of day length, but the effect of this mutation on abaxial trichome production is only observed under short days. This is common for many genes affecting vegetative phase change. For example, over-expression of miR156 or loss-of-function mutations in SPL genes have much more significant effects on abaxial trichome production in short days than in long days. Presumably this is because photoperiod promotes abaxial trichome production independently of miR156 and the SPL genes.

We have changed “chlrophyllide a oxygenase” to “chlorophyll a oxygenase”, and provide the AGI number for CH1. We note that it is standard practice in Arabidopsis genetics to use the original name for a gene. AtCAO was originally identified as CHLORINA1, so we use the original name for this gene.

*5) Differences in glucose concentrations used*.

It is our understanding that Jia-Wei performed most experiments using 50 mM glucose. We had originally tested concentrations of glucose up to 80 mM, and observed no greater effect on miR156 abundance than with 10 mM glucose. We have now included these data in Figure 3C. We used 10 mM glucose for all our experiments because high concentrations of glucose have deleterious effects on plant growth.

*Comments specific to your paper*:

*A) Different sugar concentrations were used for the gene expression and promoter experiments; for the former 0.5–20 mM glucose is used, while 1% (∼50 mM) was used for the latter. Is there a reason for this? If so, please make that clear to the reader*.

As noted above, we have re-examined the effect of sugar on the expression of these reporters. The GUS-fusion experiments were performed with 50 mM glucose and sucrose – a concentration that is similar to the amount used for the gene expression studies – and is the concentration used by Jia-Wei Wang for most of his experiments. 10 mM glucose or sucrose had no obvious effect on the expression of these reporters.

*B) You should attempt to quantify GUS activity for the MIR156 promoter response experiments*.

We now include qRT-PCR data for the expression of these reporters (Figure 5).

*C) Regarding the signaling role of HXK1 presented in Figure 6A and 6B, how many biological replicates were performed*?

In 10/12 independent experiments without sugar treatment, Ler accumulated more mature miR156 than *gin2-1*. In 9/10 independent experiments with sugar treatment, *gin2-1* did *not* display a reduced level of miR156 relative to wild-type plants. Experiments with the transgenic lines (*HXK1/gin2-1* and *S177A/gin2-1*) were performed 5 times, and in four cases we obtained results like those shown in Figure 6C. We have included a figure supplement to demonstrate the repeatability of these results (Figure 6–figure supplement 1).

*D) The plant growth conditions for the gin2 experiments are very different from those of the other experiments: plants were grown at 16ºC instead of 22ºC and the light intensity was much lower (60 vs 200 micromol). What is the rationale for this, especially since it is has been reported that MIR156A becomes up-regulated at lower temperatures (see e.g., Lee et al., 2010) and the differences between Ler (WT) and gin2-1 are only marginal (Figure 6C)*?

We reasoned that low temperature, low light intensity might reduce the accumulation of endogenous sugar, which would provide a sensitized background to score the *gin2-1* phenotype. The increased number of juvenile leaves (18 vs 5) under these conditions is consistent with reports (mentioned above) that low temperature increases miR156 level. We observed the same results under 23°C, SD conditions, although the difference was less significant. This latter result is now presented in Figure 6–figure supplement 2.

*E) Furthermore, are the complemented gin2 mutants significantly different from the gin2 single mutants (column 2 vs 4)? If not, this affects the ability to draw conclusions about the role of HXK1 in the regulation of miR156 levels*.

Abaxial trichome production in plants containing the complementing transgenes was not significantly different from wild-type (p > 0.1), demonstrating that these transgenes correct the phenotype of *gin2-1*. *gin2-1* mutants were significantly different from both wild type and transgenic plants.

*F) In Figure 6, it would be useful to include results for Ler + glucose when grown under the same conditions. In Figure 6A it is shown that gin2-1 w/o glucose has similar low levels of miR156 as Ler + glucose, but it would be useful to know whether at the morphological level (# of leaves w/o abaxial trichomes) the outcome is the same*.

The effect of sugar and *gin2-1* on leaf morphology is difficult to score in a Ler background for two reasons. Ler plants produce adult leaves very early on plates (at about leaves 3–4). Given that we have never observed abaxial trichomes on leaf 1 or 2 in any ecotype (including Ler), this gives a very small window (1–2 leaves) to score the *gin2-1* phenotype, greatly reducing any possible difference that may exist. Furthermore, it is difficult to score leaf shape on plants growing on plates because leaves do not develop completely normally under these conditions; this is the reason we did not do this experiment.